# 24 h Activity Guidelines in Children and Adolescents: A Prevalence Survey in Asia-Pacific Cities

**DOI:** 10.3390/ijerph20146403

**Published:** 2023-07-19

**Authors:** Phaik Ling Quah, Benny Kai Guo Loo, Sachith Mettananda, Senuri Dassanayake, Michael Yong Hwa Chia, Terence Buan Kiong Chua, Teresa Shu Zhen Tan, Poh Chong Chan, Betty Wai-Man But, Antony Chun-Cheung Fu, Shirley Man-Yee Wong, Nobuhiko Nagano, Ichiro Morioka, Shyamal Kumar, Muttathu K. C. Nair, Kok Hian Tan

**Affiliations:** 1Division of Obstetrics & Gynaecology, KK Women’s and Children’s Hospital, Singapore 229899, Singapore; 2Sports and Exercise Medicine Service, Department of Paediatrics, KK Women’s and Children’s Hospital, Singapore 229899, Singapore; 3Department of Paediatrics, Faculty of Medicine, University of Kelaniya, Kelaniya 11600, Sri Lanka; 4Colombo North Teaching Hospital, Ragama 11010, Sri Lanka; 5Physical Education & Sports Science, National Institute of Education, Nanyang Technological University, Singapore 639798, Singapore; 6Khoo Teck Puat-National University Children’s Medical Institute, National University Hospital, Singapore 119074, Singapore; 7Department of Paediatrics, Queen Elizabeth Hospital, Kowloon, Hong Kong; 8Department of Paediatrics, Princess Margaret Hospital, Kowloon, Hong Kong; 9Department of Pediatrics and Child Health, Nihon University School of Medicine, Tokyo 173-0023, Japan; 10NIMS-Spectrum-Child Development Research Centre, Thiruvananthapuram 695123, India; 11Department of Maternal Fetal Medicine, KK Women’s and Children’s Hospital, Singapore 229899, Singapore; tan.kok.hian@singhealth.com.sg; 12Duke-NUS Medical School, Singapore 169857, Singapore

**Keywords:** 24 h activity guidelines, children, adolescents, overweight, obesity, moderate-to-vigorous physical activity, screen viewing time, sleep duration, Asia-Pacific

## Abstract

This study aimed to examine the prevalence of adherence to 24 h activity guidelines in children and adolescents from Asia-Pacific cities. In 1139 children aged 5–18 years, moderate-to-vigorous physical activity (MVPA), screen viewing time (SVT), sleep duration, child weight, height, sex, and age were parent-reported. Descriptive statistics were used to assess the number of guidelines met, and prevalence of adherence to activity guidelines by city and child sex. Prevalence of meeting all three 24 h activity guidelines was low across all countries (1.8–10.3%) (*p* < 0.05). Children from Thiruvananthapuram, India had the highest [10.3% (95% CI: 6.0–17.0)], while those from Tokyo, Japan had the lowest prevalence [1.8% (95% CI: 0.5–7.0)] of meeting all three guidelines. The highest prevalence of meeting individual MVPA, SVT and sleep guidelines was found in India [67.5% (95% CI: 58.8–75.1)], Kelaniya, Sri Lanka [63.2% (95% CI: 58.7–67.4)] and Kowloon, Hong Kong [59.4% (95% CI: 51.1–65.3)], respectively. Overall, a higher prevalence of boys met all three guidelines, compared to girls [5.9% (95% CI: 4.1–8.1) vs. 4.7% (3.1–6.6), *p* = 0.32]. The prevalence of adhering to all three activity guidelines was low in all five participating cities, with a higher proportion of boys meeting all guidelines.

## 1. Introduction

A robust body of scientific literature from meta-analysis and systematic reviews has provided evidence that modifiable movement behaviours such as sufficient physical activity (PA) [1,2], low sedentary recreational screen viewing time (SVT) [3], and sufficient sleep [4,5] have all been linked with better physical health (e.g., lower body mass index (BMI), adiposity, cardiometabolic risk). The emerging concern is that children across the Asia-Pacific region are not sufficiently active [6,7] or receiving enough sleep [8,9] and engage in excessive sedentary behaviour associated with screen time use [10]. Moreover, these unfavourable lifestyle behaviours (inadequate PA and high sedentary behaviours) often manifest during childhood, and track into adolescence and adulthood [11].

Previously, the benefits of PA, sleep, and SVT on adiposity were studied in isolation [12]. However, several national guidelines—Canada in 2016 [13], Australia in 2019 [12], and Singapore in 2021 [14]—based on an integrated 24 h movement behaviour paradigm were developed to provide recommendations for each of the three movement behaviours (PA, sedentary behaviour (SB), and sleep) across a 24 h period, suggesting that it is a combination of all three which leads to overall improved health indicators in children [14]. Consensus statements on integrated 24 h activity guidelines for children and adolescents between ages 5–18 years old were also recently developed for the Asia-Pacific region in 2021 [14] to cope with rapid urbanization, modernization, and adoption of a lifestyle resulting in reduced PA and increasing intake of calories, and the subsequent rising rates of childhood obesity [15]. Meeting the activity guidelines on movement behaviours was categorized as moderate-to-vigorous physical activity (MVPA) (≥60 min/d) and SVT (≤2 h) for children aged 5–18 years old, and sleep of 9–11 h (between ages 5–13 years old) and 8–10 h (between ages 14–18 years old) [14].

A 2020 systematic review examined the associations between meeting the 24 h Movement Guidelines with multiple health indicators, and showed that only between 4.8% and 10.8% of children, and 1.6% and 9.7% of adolescents, met all components (PA, sedentary behaviour, and sleep) of the 24 h Movement Guidelines [12]. The low adherence to these guidelines is of public health concern due to its association with a myriad of health outcomes such as adiposity, and mental, social, and cognitive health [12,16]. Uncovering the prevalence of adherence to movement guidelines might elucidate specific lifestyle behaviours which need to be targeted to increase adherence through increased health promotion.

However, studies examining the prevalence of adherence to movement behaviours in school-going children up to adolescence and childhood obesity have largely been conducted in Western [12,17,18] or high-income Asian countries only, or in an older age group of adolescent children [19]. These studies, in general, have observed lower adherence to movement behaviours in children of older age groups, especially in adolescents and girls [18], but meeting the guidelines was favourably associated with adiposity outcomes in children and youth [12]. Little is known about the prevalence of and how these movement behaviours, individually and in combination, affect the growth of children from the school-going age group, especially from middle- and high-income countries [20] in the Asia-Pacific region.

The Integrated 24 h Activity Survey on Asia-Pacific Children and Adolescents (ISAP) study aimed to examine the prevalence of adherence to the integrated 24 h activity guidelines for children and adolescents aged 5–18 years among cities in five Asia-Pacific countries [14]. The study assessed the proportion of children sampled in participating countries who met the three activity guidelines forPA, SVT, and sleep behaviour [14] individually and in combination, and determined if these proportions differed by city and child sex.

## 2. Methods

### 2.1. Study Design

The Integrated 24 h Activity Survey on Asia-Pacific Children and Adolescents (ISAP) was a cross-sectional, multi-national study conducted in cities from five different countries (Kelaniya, Sri Lanka; Kowloon, Hong Kong; Singapore; Thiruvananthapuram, India; Tokyo, Japan). Participating study sites aimed to collect a minimum of 100 survey responses from each participating study site via a convenience sampling method with a minimum of a 12-month duration, and investigate the proportion of children aged 5–18 years old sampled from each city who met the integrated 24 h activity guidelines for children and adolescents for PA, SVT, and sleep behaviour in the Asia-Pacific region [14]. The ISAP protocol was approved by the SingHealth Centralised Institutional Review Board (CIRB Ref No.: 2021/2090) ethic committee. All participants provided informed consent for participation in the ISAP study. Data were collected from April 2021 through to July 2022. In each country, a Principal Investigator (PI) was identified to take responsibility for all aspects of conducting the study at the local level and to obtain ethics approvals from their respective institutes. The local PI and research team in each country determined the most suitable approach to recruiting and consenting parents/caregivers into the study, using a convenience sampling method within their respective data collection settings (Appendix A). All data collected from the participants in this study were anonymous and did not contain any personal identifiers. All parents/caregivers who had children within the age range of 5–18 years old were eligible to participate. There were no other exclusion criteria.

### 2.2. Survey Instrument and Mode of Data Acquisition

An online 40-item Children’s Lifestyle Questionnaire was hosted on a secured survey platform called Form.Sg (https://form.gov.sg/ accessed on 1 April 2021). This survey was previously used in Singapore to gather data on children’s lifestyle behaviours, and its findings spurred the development of the Singapore Integrated 24 h Activity Guidelines for Children And Adolescents [21].

The questionnaire comprised three segments: (a) demographic characteristics of parent and child; (b) children’s PA using 9 questions adapted from the short form International Physical Activity Questionnaire (IPAQ-SF) [22] which records the activity of four intensity levels: (1) vigorous-intensity activity such as aerobics, (2) moderate-intensity activity such as leisure cycling, (3) walking, and (4) sitting; (c) children’s sleep habits (night-time sleep and day-time napping) adapted from the Child Sleep Habits Questionnaire [23]; (d) children’s recreational screen viewing time (SVT) was captured as the average time a child spends on a screen [24] (e) Parental perception of child’ health and awareness of lifestyle guidelines for children. Details of the questionnaire items have been reported previously [25]. The questionnaire was developed to assess caregiver-reported child’s PA in a week, sleep habits, SB, and SVT on a weekday and on the weekend where the caregivers were asked to recall their child’s habitual movement behaviours in a typical 7-day week. Caregivers were asked about concerns they might have about their child’s health and their awareness of current healthy lifestyle guidelines for children.

### 2.3. Survey Administration

The Children’s Lifestyle Questionnaire was administered in English to study participants in Singapore, Kowloon, and India, where participants were required to be able to read and understand the language to be eligible for study participation. The questionnaire administered in Sri Lanka and Japan was translated to the native language of the respective country. A forward translation of the questionnaire was conducted by a bilingual translator to translate the questionnaire into the country’s mother tongue. The translation was then independently back-translated (i.e., translate back from the target language into the original language) to ensure the accuracy of the translation. Finally, the back-translation and the original document were compared for inconsistencies, and when none were found, the translation was considered equivalent [26]. The questionnaire hosted on Form.SG.com was then rolled out online and was kept ‘active’ up to 15 months for data collection, and was ‘deactivated’ once data collected had ceased. Parents gave their informed consent to participate before filling out the online survey. The raw dataset of the survey was then downloaded from the Form.Sg platform.

### 2.4. Assessing Parent-Reported Physical Activity, Sleep, Sedentary Behaviour, and Screen Viewing Time

In the study, the weighted averages of time spent on each movement behaviour across all valid days were calculated, where weekend days were weighted 2/5 relative to the contribution of weekdays. The moderate physical activity (MPA) time per day, and the vigorous physical activity (VPA) time per day were summed up to derive the MVPA time per day [27]. The amount of time spent walking was captured as light physical activity (LPA). The amount of sitting time captured was used as a proxy for SB and inactivity.

### 2.5. BMI Z-Scores and Overweight/Obesity Outcomes

BMI was calculated from reported weight and height using the formula [weight (in kg)/height (in m^2^)], and sex- and age-specific body mass index (BMI) z-scores were derived using the World Child Growth Standards for 5–19 years [28]. The cut-offs for childhood overweight (OW) and obese (OB) were defined as +1SD and +2SD, respectively, above the reference distribution, and thinness was defined as −2SD as per World Health Organization (WHO) recommendations [29].

### 2.6. Parental Perception and Awareness of Child Health and Movement Behaviours

The questions used to assess parental perception of children’s health and awareness of activity guidelines for children and adolescents were previously published [25].

### 2.7. Statistical Analyses

Descriptive data were presented as mean and standard deviation for continuous data that are normally distributed, and as median and interquartile range for non-normally distributed data. Categorical data were presented as frequencies and percentages, or 95% confidence interval (95% CI). The one-way analysis of variance (ANOVA) or *t*-test for continuous variables with normal distribution, or the Kruskal–Wallis or Mann–Whitney for continuous variables with skewed distribution was used to examine the associations between BMI z-scores and movement behaviour duration (hours/day) by city and child sex. The Chi-squared test was used to examine the associations between the perception and awareness of child health and movement behaviours by city.

Meeting the activity guidelines on movement behaviour was categorized based on the recommended Asia-Pacific integrated 24 h activity guidelines for children and adolescents [15]—MVPA (≥60 min/d), SVT (≤2 h), sleep of 9–11 h (for 5–13 years old), and 8–10 h (for 14–18 years old) [14]. To assess the frequency, along with the individuals and different combinations of movement guidelines, variables used in this study were: (i) the number of the guidelines being met (from 0 = “none met” to 3 = “all three guidelines met”), and (ii) combinations of the guidelines being met as a category variable (“none”, “MVPA only”, “SVT only”, “sleep only”, “MVPA and SVT”, “MVPA and sleep”, “SVT and sleep”, and “all three”). City and child sex differences in adherence with the 24 h movement guidelines were tested using one-way ANOVA (dependent variable was the number of the guidelines being met) or the independent *t*-test, and Chi-squared test (dependent variable was combination of the guidelines being met).

All analyses were conducted using Stata version 13.1 (StataCorp LP, College Station, TX, USA). Two-sided *p*-values < 0.05 were accepted as statistically significant.

## 3. Results

The study consisted of 1139 participants. The city with the highest proportion of participants was Kelaniya, Sri Lanka (41.3%); 75.1% of the responders were mothers, 49.4% had a university degree, and they had a mean (SD) age of 39.5 (10.6) years. In this sample, 48.2% of the children were males, 61.7% were first born, with a mean (SD) age 11.3 (5.0) years, and 94.3% were free from any chronic illnesses (Table 1). Baseline characteristics stratified by city are presented in Appendix A.

### 3.1. BMI, Overweight, Obese, and Movement Behaviours, Stratified by City and Child Sex

Across the cities, mean (SD) BMI z-score was the highest in children from Singapore [(0.16 (2.8)], along with the highest prevalence of overweight 17.7% (95% CI: 13.3–17.2). Children from the cities in India had the highest prevalence of obesity 22.7% (95% CI: 15.8–31.5) and thinness [28.8% (95% CI 20.5–37.3)], while Tokyo, Japan had the lowest prevalence of obesity at 3.3% (95% CI: 1.1–9.8), and thinness at 6.6% (94% CI: 3.0–14.0) (Table 2).

Comparing between sexes, boys had a higher mean (SD) BMI z-score [−0.01 (3.0) vs. −0.23 (2.6), *p* = 0.07] compared to the girls. There was a higher prevalence of boys who were overweight [14.6% (95% CI: 11.9–18.3) vs. 13.7% (95% CI: 5.9–10.9)], and obese [14.8% (95 CI%: 11.9–18.3) vs. 7.9% (95% CI: 5.9–10.6)], compared to girls (*p* = 0.002) (Table 2).

Table 3 shows that movement behaviours significantly differed across all cities (*p* < 0.05), with children from Tokyo having the lowest; in contrast, children from India had the highest VPA, MPA, and LPA involvement. Children from Tokyo engaged in the highest duration of SVT, versus Kowloon, Hong Kong with the lowest duration. Children in Singapore spent the highest duration in both SB and on SVT, while children from the cities in India spent the least amount of time in SB. In the 5–13-year-old age group, children from both Kelaniya and India had the shortest sleep duration, while in the 14–18-year-old age group, children in Singapore had the shortest sleep duration. Children in Tokyo, Japan had the highest sleep duration for both age groups (Table 3). Between child sexes, significant differences were seen in the duration spent in VPA and SVT, with lower time spent for both activities in girls, compared to boys (*p* < 0.05) (Table 3).

### 3.2. Adherence to Integrated 24 h Activity Guidelines, Stratified by City and Child Sex

Table 4 shows the adherence to integrated 24 h activity guidelines by city and child sex. The number of the guidelines being met ranged from 0.96 (95% CI 0.85,1.06, Singapore) to 1.49 (95% CI 1.35,1.62, India) with significant differences across all cities (*p* < 0.05). Across the cities, differences were also found in the combinations of guidelines being met (*p* < 0.001), ranging from 1.8% (95% CI 0.5,7.0), Tokyo) to 10.3% (95% CI 6.0,17.0), India) meeting all three guidelines (Table 4). Cities with the highest prevalence of meeting MVPA, SVT, and sleep guidelines were India [67.5% (95% CI: 58.8–75.1)], Kelaniya [63.2% (95% CI: 58.7–67.4)], and Kowloon [59.4% (95% CI: 51.1–65.3)], respectively. Children in Kowloon also had the highest prevalence [37.5% (95% CI: 30.9–43.9) of meeting at least two of the guidelines (SVT and sleep), while children in India had the highest prevalence of meeting both MVPA and SVT [30.2% (95% CI: 22.8–38.8), as well as MVPA and sleep guidelines [21.4% (95% CI: 15.1–29.5)] (Table 4). Children in Singapore had the highest prevalence of meeting none of the activity guidelines [33.2% (95% CI: 28.0–40.0)] (Table 4).

Overall, the number of 24 h activity guidelines met was higher in boys than in girls [1.31 (95% CI 1.24, 1.38) vs. 1.21 (95% CI 1.15, 1.28), *p* = 0.05] (Table 4). A significantly higher proportion of boys met the MPVA guidelines [(29.1% (95% CI: 25.5–33.0)) vs. (21.0% (95% CI: 17.9–24.4)), *p* = 0.002], sleep guidelines [50.8% (95% CI: 46.6–54.9)) vs. 42.9% (95% CI: 36.9–48.9), *p* = 0.004], and the MVPA and sleep guidelines [13.8% (95% CI: 11.1–17.0)) vs. 8.3% (95% CI: 6.3–10.8)), *p* = 0.002], compared to the girls (Table 4). In contrast, a significantly higher proportion of girls met the SVT guidelines 57.5% (95 CI%: 53.4–61.4)) vs. 51.4% (95% CI: 47.2–55.7), *p* = 0.002], compared to the boys (Table 4).

### 3.3. Parental Perception and Awareness of Child Health and Movement Behaviours

In Appendix A, collectively across the participating cities, 10.3–14.9% of parents perceived their child to be overweight, and 4.0–16.2% believed that their child was overconsuming calories. There were 22.2–52.6% of parents who did not think their child was receiving sufficient PA, while 19.1–86.4% were not aware of PA guidelines for children. More than 50% of parents were concerned about their child’s SVT exposure, and 19.8–74.6% were not aware of SVT guidelines. Lastly, 6.2–39.6% of parents did not think their child was receiving sufficient sleep, and only 8.7–53.6% were aware of the sleep guidelines (Appendix A).

## 4. Discussion

To the best of our knowledge, this study is the first to compare the prevalence of adherence to the integrated 24 h activity guidelines in children aged 5–18 years old among different cities in the Asia-Pacific region. This study found significant differences in guideline adherence across all cities with generally low compliance (range 1.8–10.3%) in terms of meeting all three guidelines. This observation was mainly driven by a low prevalence in meeting MVPA guidelines (9.9–25.7%) in four out of five participating countries. Overall, a higher proportion of boys were meeting all three guidelines.

The observation of overall low adherence to all three movement behaviours in this age group is consistent with previously reported systematic reviews [12,18], and the overall adherence of below 25% [12]. A systematic review and meta-analysis by Tapia-Serrano et al. including 387,437 participants and 23 countries reported low adherence [3.80% (95% CI: 2.78–4.82%)] to the 24 h movement guidelines in Asian children aged 3–18 years [18]. The Asian countries included in this review were ones with high Human Development Indexes such as Singapore, China, Japan, and South Korea. This study revealed that overall guideline adherence appears to be higher for children and adolescents (5–18 years old) from the upper-middle and lower-middle-income countries, which were Sri Lanka (6.8%) and India (10.3%), respectively. The participating countries in this study ranged from lower-middle to high-income countries, which might explain the differences in the range of guideline adherence between this study and the study by Tapia-Serrano et al. [18].

The findings from this study elucidated that the overall low adherence to all three guidelines in high-income countries was mainly driven by the low prevalence in meeting the MPVA-only guidelines (9.9–21.8%). These results corroborate previously published studies in East Asian countries such as China (8.9% of children 11–16 years) [30], Hong Kong (9.5% of children 11–18 years) [31], and Korea (4.9% of children 12–18 years) [32], all of which have reported a low prevalence in meeting MVPA guidelines, leading to an overall low prevalence in meeting movement guidelines. The findings of children’s low participation in physical activity were also reflected by the higher percentage of parents concerned that their child might not be receiving sufficient physical activity, and lower percentage of parental awareness of existing physical activity guidelines, compared to other movement behaviours of sleep and SVT (Appendix A).

In this study, it was observed that children from cities in India spent the least time in sedentary behaviour (SB) (median 3.3 h/day) and had the highest prevalence of meeting the MVPA-only guideline (67.5%), compared to other countries (14.5–25.7%). The WHO report on adolescents 11–17 years old indeed highlighted that lower levels of insufficient physical activity in Asia (East, South, and Southeast Asia) were found in India [19]. In contrast, higher levels of insufficient physical activity involvement come from high-income Asia-Pacific countries such as Singapore [19,33]. The lack of physical activity involvement from other participating cities in this study could be explained by the higher leisure-time SB (range: 4.6–7.2 h/day) which is considered “high SB” [34], coupled with SVT exceeding 2 h in 36.8–58.5% of children with SVT duration ranging from 1.8–2.3 median hours/day. Cities from high-income countries such as Singapore, Japan, and Hong Kong fall in the higher range of leisure-time SB (6.0–7.2 h/day), and low PA adherence (prevalence: 9.9–21.8%). Published studies provide evidence on how parents in countries with more advanced economies are also more likely to indulge children with digital technology. Children in Hong Kong [35], Japan [36], and Singapore [37,38] were reported to be top smartphone owners in Asia with close to 100% of adolescents owning smartphones, thus creating an environment that promotes higher SB which is highly correlated with high screen time usage [39]. Among the parents from these three countries, 62.2–72.4% expressed concern regarding their children’s exposure to SVT, with 25.4–70.3% not being aware of the appropriate SVT recommendations for their child’s age group (Appendix A). Aside from high screen time use, these same countries also place a high emphasis on academic achievement and competition [40], where school-age children spend a considerable amount of time with homework assignments or tuition requirements which might be viewed by caregivers as more important than engaging in PA [7]. This academic culture, coupled with high screen time use, has undoubtedly led to higher amounts of time spent being sedentary.

Children from Singapore, and those from cities in India and Sri Lanka had lower prevalence of meeting the sleep guidelines, compared to the other cities in this study sample. The sleep duration of children from India in our study is in agreement with the findings from the multinational International Study of Childhood Obesity, Lifestyle and the Environment (ISCOLE) study of children aged 9–11 years old which reported a mean sleep duration of 8.5 h per day from a study sample in Bangalore, India [41]. This study had also found an association between low/middle income countries with lower sleep duration, compared to high-income countries such as Australia, Canada, Finland, and the United Kingdom. However, this observation does not apply to Singapore as a high-income country, and may be explained by the high SVT use (i.e., exceeding 2 h) and the low adherence to SVT guidelines, as longer screen media usage may directly reduce sleep duration by delaying or interrupting sleep time [42,43]. Furthermore, as a high-income country in East Asia, Singapore is a highly competitive society where students spend more time studying at the expense of sleep for the sake of achieving high academic performance [44,45]. These findings suggest for more sleep studies to be conducted in the Asian region to determine the differences between the sleep patterns of children and a country’s income group. Higher adherence to sleep guidelines in both Kowloon, Hong Kong and Tokyo, Japan in this study despite the high use of SVT and similar academic pressures of an Asian society may be explained by the higher percentage of children aged 5–13 years in the sample size for both countries (90% and 80%, respectively), compared to Singapore (70%). Sleep problems in both Hong Kong [46] and Japan [9] have been shown to be more prevalent specifically in the adolescent age which might not be well represented in this study’s sampling.

The study findings have also highlighted significant differences in adhering to MVPA, SVT, and sleep guidelines between boys and girls. A higher proportion of boys were meeting the MVPA and sleep guidelines, whilst a higher proportion of girls were meeting the SVT guidelines. The WHO survey study from 146 countries [19], a review in Asian school-age children [7], and a Health Behaviour in School-Aged Children (HBSC) study in 32 countries [47] have highlighted the growing concern of the gender gap in meeting physical activity recommendations between boys and girls, where boys have always been more active than girls all throughout childhood [19]. These differences in gender and meeting physical activity recommendations have also been reported in other studies using objective measures [48]. The trend of a higher proportion of boys meeting sleep guidelines but a lower proportion of them meeting SVT guidelines compared to girls in this study was also previously reported in studies from China [31], Canada [49], Europe [50], and Korea [33]. Interventions aimed at improving lifestyle behaviours amongst youth in Asian countries might consider having more resources and support targeting girls because they have been shown consistently in this study to be less active and receive less sleep.

The strength of this study lies in the wide range of sociocultural variability and various income groupings according to World Bank classifications amongst the five participating Asia-Pacific cities that allows a comparison of lifestyle behaviours of children and adolescents in the school-going age group from ages ranging from 5–18 years old in these different settings. This study had limitations which need to be addressed. Firstly, it lacked data on children’s dietary intake that could have provided insight on the total energy intake that is necessary to complete the energy balance cycle, and to assist in better elucidating the associations between the movement behaviours and BMI. For example, the children sampled from India in this study have the highest adherence to all three movement behaviours but also the highest prevalence of overweight/obese combined, while the children sampled from Japan have the lowest adherence to MVPA guidelines but the lowest prevalence of overweight/obese combined. This phenomenon can be best disentangled by understanding how energy expenditure from the movement behaviours might be affected by the energy intake. Secondly, for future similar studies, measured height and weight would provide more accurate BMI, and the use of objective measures would be better able to quantify movement behaviours accurately (physical activity, sleep, and sedentary behaviour). This will prevent misclassifications of clinical outcomes of overweight and obesity, and meeting activity guidelines recommendations, compared to the use of parent-reported data. Thirdly, since the sampling was done using a convenience method, the sample from each country was not representative of the study population. Lastly, the findings of this study may not be generalizable due to the majority of participants having received higher education (university level or higher); there was also a slightly higher percentage of females in the overall sample compared to boys.

## 5. Conclusions

Findings from the present study show that the prevalence of meeting all three guidelines were low (1.8–10.3%) in all cities in five Asia-Pacific countries driven by an overall low engagement in MVPA, particularly in the girls. Boys (6.0%) still had an overall higher prevalence of meeting all three guidelines, compared to girls (4.7%). Future research is still required to elucidate the contributing factors to low adherence and how these factors might differ among cities in the Asia-Pacific region.

## Figures and Tables

**Table 1 ijerph-20-06403-t001:** Baseline characteristics of the study participants.

Characteristics	N (%) or Mean (SD) (*n* = 1139)
Participant’s characteristics	
Education	
Primary or secondary	248 (21.8)
Post-secondary	328 (28.8)
University	563 (49.4)
Responders	
Mother	855 (75.1)
Father	279 (24.5)
Legal Guardian	5 (0.4)
Age, years	39.5 (10.6)
Cities	
India (Multiple cities)	126 (11.1)
Kelaniya, Sri Lanka	470 (41.3)
Kowloon, Hong Kong	192 (16.8)
Singapore	241 (21.2)
Tokyo, Japan	110 (9.6)
Child’s characteristics	
Birth order	
1st child	703 (61.7)
2nd child	342 (30.0)
3rd child or more	94 (8.3)
Sex	
Male	549 (48.2)
Female	590 (51.8)
Chronic illness	
No	1074 (94.3)
Yes	65 (5.7)
Age, years	11.3 (5.0)

**Table 2 ijerph-20-06403-t002:** Child BMI Z-score and child weight status, stratified by city and sex.

BMI Z-Score and Weight Status	India (*n* = 126)	Kelaniya, Sri Lanka (*n* = 470)	Kowloon, Hong Kong (*n* = 192)	Singapore (*n* = 231)	Tokyo, Japan (*n* = 110)	*p* Value	Boys (*n* = 549)	Girls (*n* = 590)	*p* Value
Child BMI -z score, mean (SD)	0.06 (4.4)	−0.34 (2.9)	−0.22 (1.9)	0.16 (2.8)	0.01 (1.2)	0.003	−0.01 (3.0)	−0.23 (2.6)	0.07
* Weight status, % (95% CI)						0.001			0.002
Normal weight	36.4 (27.8–45.8)	61.3 (56.1–66.1)	68.7 (61.5–75.1)	56.3 (49.8–62.6)	72.5 (62.4–80.8)		55.8 (51.3–60.2)	63.4 (59.0–67.4)	
Thinness	28.8 (20.5–37.3)	17.3 (13.7–21.6)	11.7 (7.8–17.4)	10.8 (7.4–15.5)	6.6 (3.0–14.0)		14.8 (11.9–18.3)	15.0 (12.3–18.6)	
Overweight	12.7 (7.7–20.4)	12.4 (9.4–16.2)	11.7 (7.7–17.4)	17.7 (13.3–17.2)	17.6 (11.0–26.9)		14.6 (11.9–18.3)	13.7 (5.9–10.9)	
Obese	22.7 (15.8–31.5)	9.1 (6.5–12.5)	7.8 (4.7–12.8)	15.2 (11.1–20.4)	3.3 (1.1–9.8)		14.8 (11.9–18.3)	7.9 (5.9–10.6)	

Missing data: BMI z-score (Kelaniya = 106, Kowloon = 13, Singapore = 10, India = 16, Tokyo = 19, Girls = 88, Boys = 76). * Overweight is defined as BMI z-score > 1SD; Obese is defined as BMI z-score > 2SD; Thinness is defined as <−2SD.

**Table 3 ijerph-20-06403-t003:** Descriptive characteristics of movement behaviours, stratified by city and sex.

Movement Behaviours (Hours/Day)	Multiple Cities, India (*n* = 126) Median (IQR)	Kelaniya, Sri Lanka (*n* = 470) Median (IQR)	Kowloon, Hong Kong (*n* = 192) Edian (IQR)	Singapore (*n* = 241) Median (IQR)	Tokyo, Japan (*n* = 110) Median (IQR)	*p* Value	Boys (*n* = 549)	Girls (*n* = 590)	*p* Value
Vigorous physical activity (VPA)	0.33 (1.0)	0.17 (0.3)	0.17 (0.3)	0.17 (0.4)	0.17 (0.4)	0.001	0.10 (0.6)	0.07 (0.3)	0.0001
Moderate physical activity (MPA)	2.50 (6.5)	0.13 (0.6)	0.17 (0.3)	0.10 (0.4)	0.17 (0.2)	0.001	0.13(0.2)	0.13 (0.5)	0.09
Light physical activity (LPA)	0.50 (1.0)	0.17 (1.0)	0.40 (0.5)	0.40 (0.6)	0.17 (0)	0.001	0.30 (0.8)	0.27 (0.6)	0.09
Sedentary behaviour (SB)	3.29 (4.0)	4.57 (4.4)	6.00 (3.5)	7.07 (4.0)	6.00 (4.0)	0.001	5.10 (4.5)	5.37 (4.7)	0.74
Screen viewing time (SVT)	2.11 (2.0)	1.79 (1.6)	1.64 (2.3)	2.29 (1.2)	2.29 (2.3)	0.001	2.00 (2.0)	1.90 (2.0)	0.03
Sleep (5–13 years old)	8.70 (1.7)	8.70 (1.3)	9.10 (1.0)	9.00 (1.4)	9.14 (1.2)	0.001	9.00 (1.3)	8.90 (1.4)	0.20
Sleep (14–18 years old)	9.00 (4.8)	8.10 (1.6)	8.10 (1.4)	7.12 (2.0)	9.10 (8.6)	0.001	8.20 (1.6)	7.90 (1.8)	0.08

VPA vigorous physical activity; MPA moderate physical activity; LPA light physical activity; SB sedentary behaviour; SVT screen viewing time.

**Table 4 ijerph-20-06403-t004:** Proportion of children meeting the 24 h movement behaviour guideline recommendation, stratified by city and sex.

Variables	Number of Guidelines Met, Mean (95% CI)	Combination of Guidelines Met (95% CI)							
		All Three (MVPA, SVT, and Sleep)	MVPA	SVT	Sleep	MVPA and SVT	MVPA and Sleep	SVT and Sleep	None
India (*n* = 126)	1.49 (1.35–1.62)	10.3 (6.0–17.0)	67.5 (58.8–75.1)	48.4 (39.8–57.1)	35.1 (26.9–44.3)	30.2 (22.8–38.8)	21.4 (15.1–29.5)	15.1 (9.8–22.5)	7.1 (3.5–13.5)
Kelaniya, Sri Lanka (*n* = 470)	1.34 (1.27–1.41)	6.8 (4.9–9.4)	25.7 (21.9–29.9)	63.2 (58.7–67.4)	45.0 (40.9–45.6)	14.0 (11.2–17.5)	13.2 (10.4–16.7)	26.3 (22.5–30.5)	12.6 (9.9–16.0)
Kowloon, Hong Kong (*n* = 192)	1.29 (1.17–140)	3.1 (1.5–7.0)	9.9 (6.4–15.0)	59.9 (52.8–66.6)	59.4 (51.1–65.3)	5.2 (2.8–9.4)	7.3 (4.5–12.4)	37.5 (30.9–43.9)	17.7 (12.9–24.1)
Singapore (*n* = 241)	0.96 (0.85–1.06)	3.4 (1.7–6.6)	14.5 (10.6–19.6)	41.5 (35.4–47.8)	40.7 (34.4–46.9)	7.1 (4.4–11.1)	6.6 (4.2–10.8)	19.5 (14.5–24.5)	33.2 (28.0–40.0)
Tokyo, Japan (*n* = 110)	1.25 (1.11–1.38)	1.8 (0.5–7.0)	21.8 (15.0–30.4)	43.6 (34.6–53.1)	59.1 (49.6–67.2)	7.3 (3.7–13.9)	5.5 (2.5–7.7)	29.1 (21.3–38.3)	15.5 (9.8–23.5)
Country differences	F = (9.95) = 4.0, *p* < 0.001	ꭓ^2^ (1, N = 1139) = 14.7, *p* = 0.012	ꭓ^2^ (5, N = 1139) = 162.3, *p* < 0.001	ꭓ^2^ (5, N = 1139) = 45.3, *p* < 0.001	ꭓ^2^ (5, N = 1139) = 38.8, *p* < 0.001	ꭓ^2^ (5, N = 1139) = 58.6, *p* < 0.001	ꭓ^2^ (5, N = 1139) = 27.2, *p* < 0.001	ꭓ^2^ (5, N = 1139) = 27.2, *p* < 0.001	ꭓ^2^ (5, N = 1139) = 58.8, *p* = 0.001
Total (*n* = 1139)	1.26 (1.21–1.31)	5.3 (4.2–6.8)	24.9 (22.3–27.3)	54.5 (51.6–57.3)	46.7 (43.9–49.8)	12.2 (10.4–14.2)	11.0 (9.30–12.9)	25.8 (23.3–28.4)	17.5 (15.4–19.7)
Boys (*n* = 549)	1.31 (1.24–1.38)	6.0 (4.3–8.3)	29.1 (25.5–33.0)	51.4 (47.2–55.7)	50.8 (46.6–54.9)	13.7 (11.0–16.8)	13.8 (11.1–17.0)	26.9 (23.3–30.6)	17.1 (14.2–20.5)
Girls (*n* = 590)	1.21 (1.15–1.28)	4.7 (3.2–6.7)	21.0 (17.9–24.4)	57.5 (53.4–61.4)	42.9 (36.9–48.9)	10.8 (8.6–13.6)	8.3 (6.3–10.8)	24.7 (21.4–28.4)	17.8 (14.9–21.1)
Sex differences	t = −1.99, *p* = 0.05	ꭓ^2^ (1, N = 1162) = 0.93, *p* = 0.32	ꭓ^2^ (1, N = 1162) = 9.55, *p* = 0.002	ꭓ^2^ (1, N = 1162) = 5.5, *p* = 0.002	ꭓ^2^ (1, N = 1162) = 8.4, *p* = 0.004	ꭓ^2^ (1, N = 1162) = 1.86, *p* = 0.17	ꭓ^2^ (1, N = 1162) = 9.48, *p* = 0.002	ꭓ^2^ (1, N = 1162) = 0.62, *p* = 0.43	ꭓ^2^ (1, N = 1162) = 0.04, *p* = 0.85

MVPA Moderate vigorous physical activity; SVT Screen viewing time. MVPA: >60 min per day; SVT: <2 h per day; Sleep duration (ages 5–13 year): 9–11 h; Sleep duration (ages 14–18 years): 8–10 h.

## Data Availability

Data described in the manuscript and analytic code will be made available upon request. Please contact the corresponding author for more information (quah.phaik.ling@kkh.com.sg).

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
