# Peer review of "24 h Activity Guidelines in Children and Adolescents: A Prevalence Survey in Asia-Pacific Cities"

_ijerph, 2023, doi:10.3390/ijerph20146403_

Round 1

Reviewer 1 Report

Dear authors,

I would like to congratulate you on the manuscript you have written. In the attached document you can find my review comments.

Kind regards

Author Response

Editor-in-Chief and reviewers

2nd July 2022

Dear Editor and reviewers,

Re: Submission of revised manuscript for evaluation and responses to reviewers’ comments and suggestions.

On behalf of the co-authors, we would like to thank the International Journal of Environmental Research and Public Health for reviewing this manuscript and the opportunity to re-submit a revised version of the manuscript entitled “"24-hour activity guidelines in children and adolescents: A prevalence survey in Asia-Pacific cities for consideration in International Journal of Environmental Research and Public Health. We greatly appreciate the constructive comments and suggestions provided by the reviewers which have helped improve the quality of the manuscript. We have addressed the reviewers’ comments as follows, and all changes made in the revised manuscript are in red text and have been underlined:

Reviewer 1:

1) The first-person plural is used to describe information. For instance, in the third line of the "study design" section, the following wording has been inserted: "we aimed to collect...". In other lines the possessive determinant "our" is also used. In scientific texts the third person singular/plural or impersonal style should be used. The entire manuscript should be revised in this respect, since this error is maintained in all sections.

Reply: The entire manuscript has been revised to maintain an impersonal style throughout the manuscript.

 2) Based on the manuscript template of the IJERPH journal the sections should be listed: 1) Introduction, 2) Methods, etc. The different sub-sections should also be listed.

Reply: The manuscript is now in the template of the IJERPH journal with sections and sub-sections listed.

3) ABSTRACT  According to the IJERPH manuscript template, the headings of the sections of the manuscript (introduction, methods, results and conclusion) should not be included.

Reply: The headings have now been removed from the abstract

4) INTRODUCTION 4) This section is very well elaborated, congratulations! Despite the congratulations, I think it is important to mention the results of an article that is currently published in IJERPH and that is related to the research topic of the manuscript. You can consult the reference in the following link: https://www.mdpi.com/1660- 4601/20/3/2004 It should also be referred to in the discussion section.

Reply: Thank you for your suggestions to include this reference. This reference has been included in the introduction section. However, I am unable to fully insert the reference in this version of the manuscript because all my references have been unlinked from my EndNote, but I will do so in the final version of the manuscript.

5) MATERIALS AND METHODS 5) In the "study design" section, it should be specified how the convenience sample was selected. For example, specify the number of centres of participants in each city, whether it was by contact with relatives, etc.

Reply: Supplementary Table 1 of this manuscript provides the details of the institutions involved in the data collection, and the settings for data collection.

Furthermore, in the manuscript it was stated that “The local PI and research team in each country determined the most suitable approach to recruiting and consenting parents/caregivers into the study, using a convenience sampling method within their respective data collection settings (Table S1). All data collected in this study was anonymous and did not contain any personal identifiers. All parents/caregivers who had children within the age range of 5-18 years old are eligible to participate. There were no other exclusion criteria.”

Thus, the Principal Investigators were responsible to decide how to proceed with data collection as long the participants met the eligibility of the study criteria.

6) RESULTS 6) I consider that the section "parental perception and awareness of child health and movement behaviors" could be deleted as it does not have much relation with the research aim. In addition, table S3 could be deleted.

Reply: The results from the section “parental perception and awareness of child health and movement behaviors" is supplementary data the assists in justifying some of the results from the study. For example, in the discussion section of paragraph 3: “The findings of children’s low participation in physical activity were also reflected by the higher percentage of parents concerned that their child might not be receiving sufficient physical activity, and lower percentage of parental awareness of existing physical activity guidelines, compared to other movement behaviours of sleep and SVT (Table S3).”

Thus, the section on parental perceptions will be maintained in this manuscript. Section 2.6 of the methods section have also elaborated that the set of questions used to assess parental perception and awareness of child health and movement behaviors have been previously published in “Quah, P.L., et al., Parental perception and guideline awareness of children's lifestyle behaviours at ages 5 to 14 in Singapore. Ann Acad Med Singap, 2021. 50(9): p. 695-702.”

7) CONCLUSIONS 8) This section should be expanded based on the aim of the research and the results obtained. I would like you to take into consideration all the comments made after the revision of the manuscript.

Reply: The overall findings of the study highlight that the adherence to all movement behaviours were low in all the cities that participated in this study, and this was mainly driven by low engagement in physical activity. The differences between genders contributed to this observation.

The conclusion has been revised to: “Findings from the present study show that the prevalence of meeting all three guidelines were low (1.8%-10.3%) in all cities in five Asia-Pacific countries driven by an overall low engagement in MVPA, particularly in the girls. Boys (6.0%) still had an overall higher prevalence of meeting all three guidelines, compared to girls (4.7%). Future research is still required to elucidate the contributing factors to low adherence and how these factors might differ among the cities in Asia-Pacific.

Reviewer 2 Report

Dear Authors,

Thank you for the opportunity to read your paper. This is an interesting, collaborative effort examining children’s movement behaviours and I applaud the team on this work. I believe this could be a good contribution to the literature on Asian Pacific children’s movement behaviours; however, some edits could strength the current manuscript. You can find my comments below. Best of luck with your revisions!

Page 2, Paragraph 1: expand on the sentence, “Moreover, these unfavourable lifestyle behaviours, that often manifest during childhood and track into adolescence and adulthood”. It seems like the second part of the sentence is missing.

Page 2, Paragraph 2: Based on the first paragraph, it seems like you are taking a holistic approach to thinking about all facets of health; however, this paragraph is focused on adiposity. Potentially ‘movement behaviours’ may be more appropriate in this circumstance. I would recommend starting the manuscript with the impact of adiposity on children’s health if that is the targeted health outcome.

Page 2, Paragraph 3: what do the two different percentages refer to? Boys versus girls? Based on the current sentence structure it seems that there should be only one statistic.

Page 3, Paragraph 5: Were there a certain number of days of data that needed to be collected to be included in the study? What counted as a valid day?

Page 4, Paragraph 2: age is associated with movement behaviours and adherence to the guidelines as a result. Throughout the literature, we have seen that there are declines in physical activity from childhood (aged 5-12) to adolescence (aged 13-18). If you have access to this information, this would be beneficial to add to your analysis. 

Page 4, Paragraph 3: Information on the movement guidelines should be presented earlier in the manuscript. This could either be in the introduction when you discuss the introduction of the movement guidelines or the study design in the methods.

Page 7, Paragraph 1: How did you analyze parents’ perceptions? This seems to be missing from the statistical analysis section of the methods.

Page 8 + 9: You have some interesting findings in this paper related to BMI, but these are not included in the discussion. For instance, boys had a higher BMI yet higher PA. Tokyo has the lowest movement behaviours but the lowest BMI, while India has the highest physical activity levels but the highest prevalence of obesity. Can you explain these relationships in your discussion?

Page 8, Paragraph 2 + 3: Table S6 doesn’t exist in your additional files.

Page 8: some of your statistics are not followed by a reference. Please double-check and add references where appropriate.

Page 9, Paragraph 3: Need to add the limitations of your study. As you used convenience sampling to collect participants, was the sample representative of your study population? There seems to be a higher than average female, educated group. Also, as this data was collected over a long period of time, did you account for factors like seasonality and activity availability/competitive seasons? You were also working with a variety of institutions across multiple counties. Were there any issues with the consistency or quality of delivery?

Author Response

Reviewer 2

  • Page 2, Paragraph 1: expand on the sentence, “Moreover, these unfavourable lifestyle behaviours, that often manifest during childhood and track into adolescence and adulthood”. It seems like the second part of the sentence is missing.

Reply: This sentence has been revised to read as: Moreover, these unfavourable lifestyle behaviours (inadequate physical activity and high sedentary behaviours), often manifest during childhood, and track into adolescence and adulthood 1.”

  • Page 2, Paragraph 2: Based on the first paragraph, it seems like you are taking a holistic approach to thinking about all facets of health; however, this paragraph is focused on adiposity. Potentially ‘movement behaviours’ may be more appropriate in this circumstance. I would recommend starting the manuscript with the impact of adiposity on children’s health if that is the targeted health outcome.

Reply: The text in the first paragraph of the manuscript has now been amended to focus only on the impact of movement behaviours on adiposity.

A robust body of scientific literature from meta-analysis and systematic reviews have provided evidence that modifiable movement behaviours such as sufficient physical activity (PA), [1, 2] low sedentary recreational screen viewing time (SVT) [3], and sufficient sleep [4, 5] have all been linked with better, physical health (e.g., lower body mass index (BMI), adiposity, cardiometabolic risk).”

  • Page 2, Paragraph 3: what do the two different percentages refer to? Boys versus girls? Based on the current sentence structure it seems that there should be only one statistic.

Reply: There was a word “between” missing from this sentence. The sentence has now been amended.

A 2020 systematic review examined the associations between meeting the 24-Hour Movement Guidelines with multiple health indicators, and showed only between 4.8% and 10.8% of children, and 1.6% and 9.7% of adolescents met all components (physical activity, sedentary behaviour and sleep) of the 24-Hour Movement Guidelines [14].”

  • Page 3, Paragraph 5: Were there a certain number of days of data that needed to be collected to be included in the study? What counted as a valid day?

Reply: No. There were no specific number of days the data had to be collected to be included in the study. The questionnaire provided to the caregivers required them to recall and report their child’s usual movement behaviours over the past 7 days. The participating study sites were given up to 12 months to achieve at least 100 recruited study participants.

  • Page 4, Paragraph 2: age is associated with movement behaviours and adherence to the guidelines as a result. Throughout the literature, we have seen that there are declines in physical activity from childhood (aged 5-12) to adolescence (aged 13-18). If you have access to this information, this would be beneficial to add to your analysis. 

Reply: Thank you for the suggestion. You are indeed accurate to point out the affect of age on physical activity engagement, and we do have the data analysis to show this trend however we plan to publish this as a separate manuscript. This manuscript just aims to describe the overall physical activity in this group of participants aged 5-18 years without stratifying by age group.

  • Page 4, Paragraph 3: Information on the movement guidelines should be presented earlier in the manuscript. This could either be in the introduction when you discuss the introduction of the movement guidelines or the study design in the methods.

Reply: The movement guidelines have now been incorporated into paragraph 2 of the introduction, and in the methods section under “statistical analysis”.

In paragraph 2 of the introduction: Meeting the activity guidelines on movement behaviours were categorized as moderate-to-vigorous physical activity (MVPA) (> 60 minutes/d) and SVT (<2 hours) for children aged 5-18 years old, and sleep of 9–11 hours (between ages 5–13 years old) and 8–10 hours (between ages 14–18 years old).

In the statistical analysis section of the methods: Meeting the activity guidelines on movement behaviour were categorized based on the recommended Asia-Pacific integrated 24-hour activity guidelines for children and adolescents [15] – MVPA (> 60 minutes/d), SVT (<2 hours), sleep of 9–11 hours (for 5–13 years old) and 8–10 hours (for 14–18 years old).”

  • Page 7, Paragraph 1: How did you analyze parents’ perceptions? This seems to be missing from the statistical analysis section of the methods.

Reply: We have now incorporated a Section 2.6 in the methods section to elaborate that the set of questions used to assess parental perception and awareness of child health and movement behaviors have been previously published in “Quah, P.L., et al., Parental perception and guideline awareness of children's lifestyle behaviours at ages 5 to 14 in Singapore. Ann Acad Med Singap, 2021. 50(9): p. 695-702.”

In Section 2.7 of the methods section, I have added on the methods used to analyze the data: “The Chi-squared test was used to examine the associations between the perception and awareness of child health and movement behaviours by city.”

  • Page 8 + 9: You have some interesting findings in this paper related to BMI, but these are not included in the discussion. For instance, boys had a higher BMI yet higher PA. Tokyo has the lowest movement behaviours but the lowest BMI, while India has the highest physical activity levels but the highest prevalence of obesity. Can you explain these relationships in your discussion?

Reply: We agree that these observations are interesting, but it is difficult to explain with the existing dataset alone. One of the potential differences contributing to these observations is the differences in energy intake or dietary habits of the children from each participating country due to the cultural differences in food intake.

These limitations have now been addressed in the final paragraph of the discussion: This study had limitations which need to be addressed. Firstly, it lacked data on children’s dietary intake that could have provided insight on the total energy intake that is necessary to complete the energy balance cycle, and to assist in better elucidating the associations between the movement behaviors and BMI. For example, the children sampled from India in this study have the highest adherence to all three movement behaviours but also the highest prevalence of overweight/obese combined, while the children sampled from Japan have the lowest adherence to MVPA guidelines but the lowest prevalence of overweight/obese combine. This phenomenon can be best disentangled by understanding how energy expenditure from the movement behaviours might be affected by the energy intake. Secondly, for future similar studies, measured height and weight would provide more accurate BMI, and the use of objective measures would be be tter able to quantify movement behaviours accurately (physical activity, sleep and sedentary behaviour). This will prevent misclassifications of clinical outcomes of overweight and obesity, and meeting activity guidelines recommendation’s, compared to the use of parent-report data.

  • Page 8, Paragraph 2 + 3: Table S6 doesn’t exist in your additional files.

            Reply: This error has now been rectified. Table S6 is meant to be Table S3 in the manuscript.

  • Page 8: some of your statistics are not followed by a reference. Please double-check and add references where appropriate.

Reply: The statistical results presented in the manuscript has now been referenced to it’s corresponding Table number.

  • Page 9, Paragraph 3: Need to add the limitations of your study. As you used convenience sampling to collect participants, was the sample representative of your study population? There seems to be a higher-than-average female, educated group. Also, as this data was collected over a long period of time, did you account for factors like seasonality and activity availability/competitive seasons? You were also working with a variety of institutions across multiple counties. Were there any issues with the consistency or quality of delivery?

Reply: We have noted these limitations as mentioned and will include it in the final paragraph of the discussion.

Other than that, there were no issues with the consistency or quality of the delivery of the results since all the data collection was via an online platform which was monitored by the coordinating center at KK Children’s and Women’s Hospital, Singapore. The only issue was the difficulty in data collection in countries such as India and Japan due to the Covid-19 restrictions between 2021- early 2022 especially in the hospital settings which slowed down the data collection and reduced the number of participants recruited from these study sites. We did not make any adjustments to account for seasonality (i.e., spring, summer, autumn and winter) but have just chosen to present the descriptives of the durations of activity in this manuscript. Furthermore, the recommendations for the Asia-Pacific integrated 24-hour activity guidelines for children and adolescents for all movement behaviours are applicable throughout the year despite the change in seasons. We did not collect any data on the activity/ competitive seasons, so we were unable to account for these factors.

The limitations section in the last paragraph of the discussion section has been expanded: This study had limitations which need to be addressed. Firstly, it lacked data on children’s dietary intake could have provided insight on the total energy intake that is necessary to complete the energy balance cycle, and to assist in better elucidating the associations between the movement behaviors and BMI. For example, the children sampled from India in this study have the highest adherence to all three movement behaviours but also the highest prevalence of overweight/obese combined, while the children sampled from Japan have the lowest adherence to MVPA guidelines but the lowest prevalence of overweight/obese combine. This phenomenon can be best disentangled by understanding how energy expenditure from the movement behaviours might be affected by the energy intake. Secondly, for future similar studies, measured height and weight would provide more accurate BMI, and the use of objective measures would be better able to quantify movement behaviours accurately (physical activity, sleep and sedentary behaviour). This will prevent misclassifications of clinical outcomes of overweight and obesity, and meeting activity guidelines recommendation’s, compared to the use of parent-report data. Thirdly, since the sampling was done using a convenience method, the sample from each country was not representative of the study population. Lastly, the findings of this study may not be generalizable due to the majority of participants being those who had received higher education (university level or higher) and there was a slightly higher percentage of females in the overall sample compared to boys.”

Reviewer 3 Report

Great job.

However, I have a question about the selection criteria for research subjects:

Why choose Kelaniya, Sri Lanka; Kowloon, Hong Kong; Singapore; Thiruvananthapuram, India; and Tokyo, Japan as the research locations? It should be noted that Hong Kong (China), Singapore, and Tokyo are typical developed regions within their respective countries or regions (China, Singapore, and Japan).

Can Kelaniya and Thiruvananthapuram serve as cities of equal status for their respective countries, Sri Lanka and India?

Please pay attention to some minor mistakes, such as:

The incorrect unit writing template: "weight (in kg)/height (in m2)."

The statistical error in Table 1: The sum of participants in education is 1138."

Author Response

Editor-in-Chief and reviewers

2nd July 2022

Dear Editor and reviewers,

Re: Submission of revised manuscript for evaluation and responses to reviewers’ comments and suggestions.

On behalf of the co-authors, we would like to thank the International Journal of Environmental Research and Public Health for reviewing this manuscript and the opportunity to re-submit a revised version of the manuscript entitled “"24-hour activity guidelines in children and adolescents: A prevalence survey in Asia-Pacific cities for consideration in International Journal of Environmental Research and Public Health. We greatly appreciate the constructive comments and suggestions provided by the reviewers which have helped improve the quality of the manuscript:

Reviewer 3

1) Why choose Kelaniya, Sri Lanka; Kowloon, Hong Kong; Singapore; Thiruvananthapuram, India; and Tokyo, Japan as the research locations? It should be noted that Hong Kong (China), Singapore, and Tokyo are typical developed regions within their respective countries or regions (China, Singapore, and Japan).

Reply: These participating countries were part of our Asia Pacific consensus workgroup which participated in the Asia-Pacific integrated 24-hour activity guidelines for children and adolescents. While creating these consensus guidelines for the Asia-Pacific countries we also proposed an Asia-Pacific wide data collection of children’s movement behaviours. The initial study involved more countries from the Asia-Pacific region such as Vietnam and Australia, but due to the poor data collection (i.e., from the participating study site in Vietnam) and study withdrawal (i.e., from the Australian study site) these study sites were not included into the final manuscript. Furthermore, since there was no funding to support a more nation-wide data collection method, it was only logistically possible to conduct the data collection from specific cities where the Principal Investigator’s local institutions were located.

2) Can Kelaniya and Thiruvananthapuram serve as cities of equal status for their respective countries, Sri Lanka and India?

Reply: No, these two cities are not representative of the entire population of Sri Lanka or India. Again, since this study was not funded, we could only conduct data collection within the cities of each study sites Principal Investigators local institutions.

These limitations are now addressed together with the other study limitations in the revised version of the manuscript in the last paragraph of the discussion: This study had limitations which need to be addressed. Firstly, it lacked data on children’s dietary intake could have provided insight on the total energy intake that is necessary to complete the energy balance cycle, and to assist in better elucidating the associations between the movement behaviors and BMI. For example, the children sampled from India in this study have the highest adherence to all three movement behaviours but also the highest prevalence of overweight/obese combined, while the children sampled from Japan have the lowest adherence to MVPA guidelines but the lowest prevalence of overweight/obese combine. This phenomenon can be best disentangled by understanding how energy expenditure from the movement behaviours might be affected by the energy intake. Secondly, for future similar studies, measured height and weight would provide more accurate BMI, and the use of objective measures would be better able to quantify movement behaviours accurately (physical activity, sleep and sedentary behaviour). This will prevent misclassifications of clinical outcomes of overweight and obesity, and meeting activity guidelines recommendation’s, compared to the use of parent-report data. Thirdly, since the sampling was done using a convenience method, the sample from each country was not representative of the study population. Lastly, the findings of this study may not be generalizable due to the majority of participants being those who had received higher education (university level or higher) and there was a slightly higher percentage of females in the overall sample compared to boys.

 3) Please pay attention to some minor mistakes, such as: The incorrect unit writing template: "weight (in kg)/height (in m2)." The statistical error in Table 1: The sum of participants in education is 1138."

Reply: Thank you for pointing out these errors. The necessary amendments have been made to the revised version of the manuscript to correct for these errors.

Round 2

Reviewer 2 Report

Dear authors, 

Thank you for the thoughtful responses to the reviewers' comments. After reviewing the manuscript, I believe the changes you made have strengthened your work. I only have one remaining comment to prepare this manuscript for publication. Congratulations on this work! 

Page 3, Paragraph 5: Were there a certain number of days of data that needed to be collected to be included in the study? What counted as a valid day?

Reply: No. There were no specific number of days the data had to be collected to be included in the study. The questionnaire provided to the caregivers required them to recall and report their child’s usual movement behaviours over the past 7 days. The participating study sites were given up to 12 months to achieve at least 100 recruited study participants.

Reviewer response: Can you specify this in your survey instrument or survey administration sections?

Author Response

Dear Reviewer 3, 

Thank you for your constructive comments which have helped improve the quality of this manuscript. 

The reply to your question is as below: 

1) Page 3, Paragraph 5: Were there a certain number of days of data that needed to be collected to be included in the study? What counted as a valid day?

Reply: No. There were no specific number of days the data had to be collected to be included in the study. The questionnaire provided to the caregivers required them to recall and report their child’s usual movement behaviours over the past 7 days. The participating study sites were given up to 12 months to achieve at least 100 recruited study participants.

Reviewer response: Can you specify this in your survey instrument or survey administration sections?

Reply: These points have now been stated in the methods section of the manuscript. All the changes made have been highlighted in red. 

In Section 2.5 Paragraph one: "Participating study sites aimed to collect a minimum of 100 survey responses from each participating study site via a convenience sampling method in a minimum of a 12-month duration, and investigate the proportion of children aged 5-18 years old sampled from each city who met the integrated 24-hour activity guidelines for children and adolescents for physical activity, screen time and sleep behaviour in the Asia Pacific region [14]. "

In section 2.2 paragraph 2: " The questionnaire was developed to assess caregiver-reported child’s physical activity in a week, sleep habits, SB and SVT on a weekday and on the weekend where the caregivers were asked to recall their child’s habitual movement behaviours in a typical 7-day week."

Reviewer 3 Report

No more questions.

Author Response

Thank you very much for your constructive comments which have improved the quality of this manuscript.